



# Measurement report: Aerosol hygroscopic properties extended to 600 nm in the urban environment

Chuanyang Shen[1], Gang Zhao[1,2], Weilun Zhao[1], Ping Tian[3], Chunsheng Zhao[1*]

[1] Department of Atmospheric and Oceanic Sciences,School of Physics, Peking University, Beijing 100871, China

[2] College of Environmental Sciences and Engineering, Peking University, Beijing 100871, China

[3] Beijing Key Laboratory of Cloud, Precipitation and Atmospheric Water Resources, Beijing 100089, China

*Correspondence to*: Chunsheng Zhao (zcs@pku.edu.cn)

**Abstract.** Submicron particles larger than 300 nm dominate the aerosol light extinction and mass concentration in the atmosphere. The water uptake ability of this size range greatly influences the particle mass, visibility degradation and particle chemistry. However, most of previous field measurements on aerosol hygroscopicity are limited within 350 nm. In this study, the size-resolved aerosol hygroscopic properties over extended size range (50-600 nm) at 85% relative humidity were investigated in Beijing winter from 27 November 2019 to 14 January 2020 using a Humidity Tandem Differential Mobility Analyser (HTDMA) instrument. The corresponding aerosol optical properties were also analyzed using the Mie scattering theory. Results show that the averaged probability distribution of GF (GF-PDF) is generally a constitute of a more-hygroscopic (MH) group and a less-hygroscopic (LH) group (including hydrophobic). For the particles larger than 300 nm, there exist a large fraction of LH group particles, resulting in an unexpected low hygroscopicity. During the development of pollution when particles are gradually aged and accumulated, the bulk hygroscopicity above 300 nm is enhanced significantly by the growth and expansion of MH group. This result is supported by previous chemical composition analysis and we give a more direct and detailed evidence from growth factor and mixing state aspects. Our calculations indicate that the optical contribution of particles larger than 300 nm constitutes about 2/3 of the total aerosol extinction. The large hygroscopic variation of aerosols above 300 nm will influence the light degradation comparably with the increase of aerosol loading in the low visibility haze events. Our studies highlight that the hygroscopic properties above 300 nm are complex and vary greatly with different pollution stages, therefore more field measurements and investigations need to be done in the future.

## 1 Introduction

Aerosol particles can scatter and absorb solar radiation, and act as cloud condensation nuclei in the atmosphere. Through these effects, they can influence the radiative balance of the Earth-Atmosphere system (ALBRECHT, 1989; CHARLSON et al., 1992; Haywood & Boucher, 2000; Lohmann & Feichter, 2005; Penner, Hegg, & Leaitch, 2001; Twomey, 1974). They can also impact human life by degrading environment visibility and despairing respiratory health (Chang, Song, & Liu, 2009; Ibald-Mulli, Wichmann, Kreyling, & Peters, 2002; Peters, Wichmann, Tuch, Heinrich, & Heyder, 1997). All these effects can be largely enhanced by aerosol's hygroscopic property (Kreidenweis & Asa-Awuku, 2014), which is the ability to absorb water



under an elevated relative humidity (RH). Particle's water uptake property, which is mainly related to the water-soluble materials contained, determines aerosol liquid water content, affects the multiphase chemistry and local photochemistry, and facilitates particle formation and aging processes (Wu et al., 2018). Thus, a thorough understanding of aerosol hygroscopicity is crucial to quantify all these effects.


 So far, while many field measurements on aerosol hygroscopicity have been carried out worldwide using Hygroscopicity Tandem Differential Mobility Analyzer (HTDMA) (Hersey et al., 2011; Kitamori, Mochida, & Kawamura, 2009; Martin et al., 2011; Massling et al., 2009; Park, Kim, & Park, 2009), Cloud Condensation Nuclei Counter (CCNC)(Deng et al., 2013; Nan Ma et al., 2016; Tao, Zhao, Ma, & Kuang, 2018), and optical humidified measurements (f(RH)) (Carrico et al., 2000;

Fitzgerald, Hoppel, & Vietti, 1982; Y. Kuang, Zhao, Tao, & Ma, 2015; Yan et al., 2009; Zieger et al., 2010), detailed hygroscopic properties over large sizes (> 300 nm) are still very scarce. CCNC can give an estimate of bulk aerosol hygroscopicity under different supersaturations. Because of the supersaturation limits, it mostly reveals hygroscopicity property within 200 nm and cannot provide direct size-resolved or mixing state information (Roberts & Nenes, 2005). F(RH) measures the humidity-dependent scattering coefficient over the whole size range. It focuses more on the hygroscopicity of

sizes that contribute more to the bulk optical properties (Y. Kuang et al., 2018), and cannot give size information or mixing states, either. HTDMA is a widely used instrument to obtain the size-resolved growth factor of aerosol particles under different RH, and has been applied to many field measurements to obtain aerosol bulk hygroscopicity as well as mixing state information. However, most of these field studies were limited within the size range of 300 nm (Swietlicki et al., 2017).

For the submicron aerosols in the urban environment, especially areas suffering from severe haze pollution and affected by intensive anthropogenic activities, the particles larger than 300 nm contribute significantly to the particle mass and surface concentration (Lundgren & Paulus, 2012; Sverdrup, 1977), liquid water content (Bian, Zhao, Ma, Chen, & Xu, 2014), and optical properties (Y. Kuang et al., 2018; Ouimette & Flagan, 1982). For the submicron particles, although the number distribution is dominated by particles smaller than 0.1 μm, most of the surface area is in the 0.1-0.5 μm size range (Seinfeld

& Pandis, 2006). The aerosol mass distribution peaks at an even higher size range of about 0.2-1 μm. Moreover, light scattering is usually approximately proportional to aerosol volume or mass (Pinnick, Jennings, & Chýlek, 1980), which means that the light degradation by aerosol particles is also concentrated at a relatively higher accumulation size. N. Ma et al. (2012) demonstrated that the aerosol particles between 200 nm and 1 um usually contribute more than 80% to light extinction at 550 nm during summer on the North China Plain (NCP). In the pollution period, the enhanced particle growth by coagulation and

condensation of vapors, aerosol size distribution will shift to larger diameters. The aerosol mass and optical extinction contributed by the larger accumulation mode particles will also be increased. When exposed to a high RH, these factors can be further enhanced with the addition of water. Considering the dominant contribution of this part to aerosol optical properties, it's necessary to study the detailed hygroscopic properties over extended size range and investigate its variation during different pollution conditions.




Some previous studies have tried to derive the hygroscopicity in a larger accumulation size above 300 nm. However, there is no direct evidence and measurements supporting their assumptions or parameterization schemes. Ye Kuang et al. (2017) calculated the equivalent aerosol hygroscopicity parameter κ from f(RH) curves ($\kappa_{f(RH)}$) measured at Wangdu on the NCP and the results lied between 0.06 and 0.51, presenting a very large variation range. Because the light scattering is mostly

contributed by particles at a larger accumulation size, so $\kappa_{f(RH)}$ can generally represent the hygroscopicity in this size range. H. J. Liu et al. (2014) also derived the hygroscopicity over a larger size range of 30 nm- 4 μm from chemical composition and found that κ values vary relatively less for particles between 250 nm and 1 μm than particles smaller than 250 nm. Chen et al. (2012) assumed that particles in each aerosol mode had the same hygroscopicity, and then estimated size-resolved κ for aerosols with diameters of 3 nm ~10 μm based on the contribution of each mode to a specific particle size. However, all these

estimations are indirect methods and have some unproven assumptions.

Beijing, one of the biggest cities in China and the world, is a densely populated area with severe particle pollution. Representative of the urban environment, the consumption of fossil fuels is quite considerable, and a lot of pollutants are emitted into the environment every day. Exacerbated by meteorological conditions, these pollutants are readily trapped in local

region, go through a series of physical and chemical processes, and finally evolved into severe haze events (Guo et al., 2014; Jiang, Wang, Zhao, Li, & Che, 2015; Sun et al., 2014; Ye, Song, Cai, & Zhang, 2016). Influenced by the particle pollution and strong hygroscopic growth of aerosol particles, visibility degradation is also a frequent and urgent environmental problem. Based on this condition, many field measurements focusing on the particles' microphysical properties including hygroscopicity were conducted in Beijing and the surrounding region. Massling et al. (2009) used a custom made HTDMA to measure the

hygroscopic properties of ambient particles in Beijing and reported the corresponding measurement results. Haze in China (HaChi) campaign also investigate aerosol particles' various microphysical properties including CCN activity, optical enhancement factor and hygroscopic growth properties under high humidity (Chen, Zhao, Ma, & Yan, 2014; Deng et al., 2011; H. J. Liu et al., 2014; P. F. Liu, Zhao, Göbel, et al., 2011). Wang et al. (2018) made statistical analysis and proposed parameterization of the hygroscopic growth of urban aerosol (50-350 nm) in Beijing from long-term measurement. However,

there are no direct measurements focusing on the size range larger than 350 nm.

In this study, we deployed an HTDMA instrument to our laboratory in Beijing to conduct aerosol hygroscopicity measurements from 27 November 2019 to 14 January 2020. Other particle microphysical properties, optical properties, and meteorology parameters were also monitored simultaneously. Three questions need to be answered here: 1) what are the basic characteristics

of aerosol hygroscopicity over extended size range in the urban environment?  2) how the hygroscopic properties vary with pollution conditions and 3) how the variation influences the visibility degradation?  In the following, section 2 will give a description of the data and methods used in this study; section 3 will present the measurement results including overall aerosol



hygroscopicity and its daily variations; the hygroscopic variation under different pollution conditions and potential impacts will be shown in section 4 and the last section comes the conclusions.

## 2. Data and methods

### 2.1 measurement site

The measurement was conducted in the aerosol laboratory, which is located on the rooftop of a six-floor building in the campus of Peking University. It shares the same location with the AERONET station of BEIJING_PKU (39˚59' N, 116˚18' E). The sampling site is in the northwest of Beijing, surrounded by schools, residential buildings and shopping centers. Moreover, two main streets: Zhongguancun North Street is located to the west and Chengfu Road is to the south. Except for the road traffic, there are no large emission sources in the neighboring. Aerosol particles here are representative of the urban environments. More details can be found in Zhao et al. (2018).

In the measurement, a PM10 impactor was used to remove aerosol particles with aerodynamic diameters larger than 10 μm. Then a dryer was used to decrease the RH to less than 30%. Next, the dried poly-disperse particles were guided into a splitter with different instruments located on the downstream. From 27 November 2019 to 14 January 2020, aerosol number size distributions, size-resolved hygroscopicity, and optical properties including scattering and absorption coefficients were measured. Meteorological parameters like temperature, wind speed, wind direction, and RH were also monitored in this site.

### 2.2 Instrumentation and data

A BMI Humidified Tandem Differential Mobility Analyser (BMI HTDMA, Model 3100) was mainly used in this study. The detailed instrument description and performance evaluation has been introduced in the previous publication (Lopez-Yglesias, Yeung, Dey, Brechtel, & Chan, 2014). Similar to other conventional HTDMA systems, this instrument mainly incorporates an upstream Differential Mobility Analyzer (DMA1), a humidification system that offers humidified sheath air to the humidifier and downstream DMA to expose the particles to a prescribed RH, and a downstream DMA (DMA2) in series with a Mixing Condensation Particle Counter (BMI MCPC, Model 1720) to measure the particles' size distribution after water uptake. Unique features of this BMI HTDMA include a diffusion-based particle humidifier, a DMA design allowing selection of particles up to 2 μm diameter at only 5600 volts, and the ability to study the complete deliquescence and efflorescence cycle. It also offers a processing software to do the corrections and data inversion, which transforms the raw measured counts to the particle's growth factor distribution.



During the measurement, the RH of the second DMA was set to be 85%. The dry diameters selected by DMA1 were 50 nm, 100 nm, 200 nm, 300 nm, 400 nm, 500 nm, 600 nm. To calibrate the measurement system, ammonium sulfate particles were also tested to compare with the theoretical values. The calibration includes both the dry test and RH test. The temperature and RH in the humidifier and DMA2 sheath were recorded real-time for later check of system stability. For the best working performance, the room was air-conditioned at 25 ℃ and circulated all the time.

The hygroscopic diameter growth factor (GF) is the ratio of the particle wet diameter at a given RH to the dry diameter, $D_0$:

$$GF = \frac{Dp(RH)}{D_0}$$

Growth factor probability distribution function (GF-PDF, c(GF)) is normalized to unity. In order to simplify parameterization and better compare the aerosol hygroscopic properties among different measurements, the hygroscopicity parameter κ was also calculated in this study (Petters & Kreidenweis, 2007). It can be calculated from the following equation:

$$\kappa = (GF^3 - 1) \left( \frac{1}{RH} \exp(\frac{4\sigma_{s/a}M_w}{RT\rho_w D_0 GF}) - 1 \right)$$

where $\sigma_{s/a}$ is the droplet surface tension, $M_w$ is the molecular weight of water, $\rho_w$ is the density of liquid water, R is the universal gas constant, and T is the absolute temperature.

The κ-PDF can also be derived from the GF-PDF. The volume-weighted mean growth factor is defined as the 3rd-moment mean values of the GF-PDF: $GF_{mean} = (GF^3 \cdot c(GF)dGF)^{1/3}$ .The corresponding mean hygroscopic parameter $\kappa_{mean}$ can be calculated from $GF_{mean}$ using the equation above. To better understand the mixing state of particles and group them in terms of hygroscopic properties, we classify the particles into a less-hygroscopic group with GF lower than 1.2, and a more-hygroscopic group with GF larger than 1.2. For each group, the number fraction, $GF_{mean}$, and standard deviation σ can be determined from the GF-PDF.

Apart from HTDMA, the aerosol light scattering coefficients at three wavelengths (450 nm, 550 nm, and 700 nm) were measured using a TSI 3563 nephelometer (Anderson & Ogren, 1998). The particle number size distribution (PNSD) was measured simultaneously by a BMI scanning electrical mobility sizer (BMI SEMS, Model 2100) and a TSI scanning mobility particle sizer (SMPS). The BMI SEMS measures PNSD with a time resolution of 10 min over the size range of 10-1000 nm. The SMPS has a time resolution of 5 min and covers the size from 10 to 600 nm. The black carbon (BC) mass concentrations were measured using an MA200 (Aethlabs, serial number 0083) at 1 min temporal resolution.



## 3. Results and Discussions

155 **3.1 Overview of GF and κ distribution**

The aerosol number size distributions during the whole measurement period are summarized in Fig. 1(a). We also summarize some previous studies on urban aerosol hygroscopicity from HTDMA measurements and present statistical results in Fig. 1(c). The reference can be found in Table 1. We can clearly see that in Beijing winter, the peak contribution to the surface area is 200-300 nm and to the mass is 300-400 nm. For optical light extinction coefficients at 550 nm, the peak diameter is around 160 400-500 nm. However, the focuses of previous HTDMA measurements are only around 50-300 nm. It's this dislocation that prompt us to conduct hygroscopicity measurement over extended size range and investigate the potential impacts on aerosol optical properties.

Figure 2 gives an overview of the average GF-PDF and the corresponding κ-PDF during the whole measurement period. It 165 can be seen that the mean growth factor distributions for all sizes show a distinct bi-modal pattern, consisting of a less-hygroscopic mode and a more-hygroscopic mode. For particles of 50 nm, the LH and MH modes cannot be clearly distinguished from each other. In general, particles of this size are not readily hygroscopic. For the size range between 100 nm and 400 nm, there exists a dominant MH group and a minor but distinct LH group. The peak of the MH group shifts significantly from about 1.2-1.4 to 1.4-1.6 as the size increases, indicating that the water-soluble or water-uptake materials in 170 larger sizes have a relatively stronger hygroscopicity than the smaller sizes. In contrast, the peak of the LH group shifts slightly. The number fraction of the LH group decreases with size when particles are smaller than 200 nm, whereas increases with size when size exceeds 200 nm. The spread (stand deviation) of the LH group has the same trend. For the particles larger than 400 nm, the LH group dominates the number fraction, resulting in the decrease of bulk hygroscopicity of large particles. Similar to the GF-PDF, κ-PDF also presents two modes corresponding to the LH and MH groups.

175

In general, our results are consistent with Wang et al. (2018) for the overlap region of 50-350 nm. We can also see a stronger fraction of nearly-hydrophobic group with size in lower and moderate pollution conditions in their study. This trend is captured by our results and further extended to a larger size above 350 nm. Massling et al. (2009) also reported the similar trend of increasing hydrophobic particles with size in winter. They also gave the size-resolved chemical composition across the 180 submicron size range. For particles larger than 300 nm, there is an increased mass ratio of Elemental Carbon and undefined mass, which may correspond to this increased group of particles. The size-resolved κ in H. J. Liu et al. (2014) are different from our results in overlap region because their measurements were done in summer, in which the emissions are quite different from that in winter. An apparent difference is the missing of nearly-hydrophobic group in accumulation mode particles, which is the main cause of the decreased hygroscopicity above 300 nm in our study.

185



The statistics of the HTDMA measurement are summarized in Table 2. The number fraction, mean GF and mean κ, spread of GF and κ for the LH and MH groups during the whole measurement period are calculated. Corresponding standard deviations are also presented in the table. The mean GF values for 50 nm, 100 nm, 200 nm, 300 nm, 400 nm, 500 nm, 600 nm are 1.13 ±0.07, 1.25 ±0.06, 1.31 ±0.07, 1.30 ±0.08, 1.24 ±0.12, 1.16 ±0.14, 1.12 ±0.16, respectively. The number fraction of LH group particles reaches a trough at 200 nm and then increases with size. The ensemble mean κ peaks at 300 nm and then decreases with size, which is related to the increase of LH group number fraction.

### 3.2. Time series and diurnal variation

The time series of aerosol size distribution and growth factor distribution over different sizes are shown in Fig. 3. The black line in panel (a) is the time-dependent volume concentrations calculated from particle number size distribution. This value can generally represent the particle pollution level. As elucidated by Guo et al. (2014), the PM episodes in Beijing present a periodic cycle governed by meteorological conditions. This kind of cycle lasts from 3 days to 1 week, causing different degrees of pollution. Our measurement captured about 6 pollution cycles. Because the sampling was done at a fixed site, so the complete cycle may go interrupted by transport or local physical or chemical processes. Nevertheless, the measured aerosol properties from 27 November 2019 to 14 January 2020 can still be representative of the typical winter conditions in Beijing. The blank area in the figure denotes missing data, which is caused by the instrument shutdown or measurement calibration.

From Fig. 3(a), we can see that the particle number concentration usually has an abrupt increase at the beginning of a pollution cycle. Then the volume or mass concentration begins to develop and increase to a high level. During the clean period, the wind is often dominated by strong northwest or northeast winds because it can bring unpolluted air masses from the northern mountainous area. Polluted days are often accompanied by weak southerly wind. Furthermore, high relative humidity tends to appear at the end of each pollution cycle, which may help to push the pollution level to a peak (Fig. 8).

For each pollution cycle, we can see from Fig. 3 (b-c) that the GF-PDF of 50 nm and 100 nm particles does not have large variations. But for other sizes, an increase of growth factor along with the pollution development can be clearly seen from the GF-PDF contour. Different sizes have different lag time to respond to the evolution of pollution: the larger the size, the more time needed to change the hygroscopic properties.

The average diurnal variations of GF-PDF for all the sizes are presented in Fig. 4. For better comparison, we mark the line between daytime and nighttime. During the measurement period, the time of sunrise was about 7.00 a.m. and the sunset was around 5.00 p.m. For particles of 50 nm, their hygroscopicity starts to decrease around sunrise and begins to increase around sunset. So the hygroscopicity parameter κ peaks at 6.00 a.m. and reaches a minimum at 19.00 p.m. For particles larger than 200 nm, the hygroscopicity at daytime are about 10-50% larger than that of nighttime.





For accumulation sizes, the number fraction of the LH group ($nf_{LH}$) in the daytime is smaller than that in the nighttime. On one hand, the cause could be that more hydrophobic particles are generated or emitted in the night, these particles include fresh BC or organic aerosols (N. Ma et al., 2011); on the other hand, more MH group particles could be formed in the daytime, or those aged particles in the night residual layer are mixed in the planetary boundary layer again (H. J. Liu et al., 2014). No diurnal variation is found in the peak of the LH group, indicating the similar chemical compositions in the LH group. In contrast, the peak hygroscopic parameter κ for the MH group varied significantly with time of a day, indicating that the chemical compositions in the MH group varied greatly at different day time. It usually reaches a maximum around noon and start to decrease in the afternoon. The black lines in Fig. 4(a-g) are the ensemble mean κ for each size. For particles larger than 50 nm, this value also peaks around noon, which may be the result of decreased $nf_{LH}$ and increased MH hygroscopicity. More statistical results about daytime and nighttime can be found in Table 3.

## 4. Evolution of Large Particle Hygroscopicity and Effects

### 4.1 Overview of GF and κ distribution

Guo et al. (2014) reported that particle pollution in Beijing is characterized by two distinct aerosol formation processes of nucleation and growth. During different aerosol formation processes, aerosol chemical compositions and corresponding hygroscopic properties may vary greatly. In order to investigate the evolution of particle hygroscopicity with different pollution levels, we group the data into three periods: 1) clean period in which the volume concentration is lower than 30 $\mu m^3/cm^3$; (2) transition period in which the particle volume concentration is larger than 30 but smaller than 40 $\mu m^3/cm^3$; (3) polluted period in which the volume concentration exceeds 40 $\mu m^3/cm^3$.

Fig.5 shows the GF-PDF and average size distribution under different pollution levels. From the clean to the transition period, the most apparent change is in the number concentration. The number concentrations of both the LH and MH mode particles have increased significantly. At the same time, the number fraction of the MH group has also been enhanced. When the air quality develops from transition to polluted level, the number concentration remains similar, but the MH group is greatly enhanced, not only in the number fraction but also in the terms of the size range. The MH mode grows to the size of 600 nm, greatly squeezing the fraction of nearly-hydrophobic mode particles. From the statistical results in Table 4, the number fraction of less hygroscopic mode for 600 nm is 0.86±0.22, 0.75±0.29, 0.39±0.27 for the clean, transition and pollution stage, respectively. For smaller particles, the variation of this value is not so distinct. Taking particles of 100 nm as an example, the $nf_{LH}$ for clean, transition and pollution stage is 0.36±0.14, 0.33±0.15, 0.30±0.13, respectively. The difference of $nf_{LH}$ among different stages increases with size.





It needs to be noted that the great decrease of $nf_{LH}$ for large particles in the pollution period doesn't necessarily mean the
decrease of absolute number concentration of LH mode particles. From Fig .5(f), the number concentration of LH mode
particles remains similar to the transition stage. However, the total number concentration and corresponding mass
concentration of large particles increases. These particles are grown from those smaller particles and are more aged in the
environment. These processes are also called secondary aerosol formation. This part of particles contributes to the increase of
the MH mode in a larger size and correspondingly total large particle number concentration. With the evolution of pollution
level, this process will continue. Zheng et al. (2016) showed that the fraction of the secondary components in PM2.5 are
enhanced with increasing PM2.5 mass concentration. Our measurement results give an another direct evidence supporting this
conclusion.

For each size, we summarize the average hygroscopicity under different pollution conditions (Fig. 6). The results show that
for particles smaller than 300 nm, the hygroscopicity variation is not very large. But for particles larger than 300 nm, particle
hygroscopicity can vary significantly with different pollution levels, which correspond to different aging stages of urban
aerosol particles. In conclusion, when urban aerosols in Beijing go through a series of aging processes, their hygroscopicity is
enhanced, especially for larger accumulation particles.

**4.2 The effects on visibility degradation**

To better understand the meteorological, microphysical and hygroscopic evolutions during the aging processes, and the effects
of hygroscopicity variation on visibility degradation, a 3-day pollution period from 27 November to 1 December 2019 is
selected for the analysis. Fig. 7 shows the overall situation of this pollution event. From the noon of 27 November, the wind
speed began to decrease to below 3 m/s. The synoptic situation of Beijing remained stagnant until the end of this pollution
period, with an average wind speed of 1.04 m/s. This meteorological situation indicated that the local emissions were the
primary contribution of the surface air pollution and the aging processes were not influenced too much from transport from
other areas. In this respect, our measurement can represent the temporal evolution of aerosols in this urban region.

We can see from Fig. 7(b-c) that in the first 24 hours, there was a considerable increase of aerosol number concentration for
particles smaller than 100 nm, which corresponded to the aerosol nucleation. The visibility dropped down from upper limit to
around 30 km. In this stage, aerosol hygroscopicity was relatively low. In the transition stage, the mean particle size grew
continuously and the aerosol volume concentration also had a stable increase. At the same time, the aerosol became more
hygroscopic in the accumulation size range between 100-300 nm. During this period, the visibility decreased slightly to 20
km. In the polluted stage, under the combined influence from the high relative humidity and large hygroscopicity enhancement,
the visibility decreased considerably to below 10 km, causing the most severe haze event, even though the aerosol volume
concentration remained similar to that in transition stage.



In this haze formation process, RH, aerosol loading (AL) and hygroscopicity all contribute to the visibility degradation. In order to quantify how much these factors took effect, we summarize the mean values of these factors in different stages, as
shown in Fig.8(a-c). We can see that as the pollution developed, the number concentration of particles larger than 200 nm increased while the small particles (<100 nm) decreased. The aerosol hygroscopicity was enhanced greatly, from a value below 0.2 to above 0.4 for particles larger than 300 nm. At the same time, the mean relative humidity also increased from 25% to 77%.

The visibility is closely related to aerosol light extinction (σ) through Koschmieder relation (Carrico, 2003; Griffing, 1980; Husar, Husar, & Martin, 2000). In this study, the particles are assumed to be spherical and their size-dependent extinction behaviour is computed from the Mie model using particle dry size distribution, refractive indices, size-resolved hygroscopicity, and ambient RH. The detailed calculation description can be found in Chen et al. (2012). From this method, we can not only quantify the contribution from each size bin, but can also determine the role each factor (AL, RH and κ) played in the low
visibility haze event. Fig. 8(d) compares the size-resolved aerosol light extinction coefficient during clean period (blue solid line) and pollution period (black solid line). The blue line is considered as the base line, which corresponds to low aerosol loading, low hygroscopicity and low RH values. When each of these factors is considered, we can obtain the following size-resolved light extinction as shown in dotted lines. The magenta dotted line represents the condition with AL effect considered, which is calculated from the PNSD in pollution stage but with the κ and RH in the clean stage. The red dotted line represents
the condition when both AL and RH are considered. The shading between lines is the difference, indicating the contribution to the aerosol light extinction from each considered factor.

From Fig. 8, we can see that particles larger than 300 nm contribute more than half of the total light extinction. Under this condition, if the hygroscopicity properties and variations in this size range are not known, it would be hard to fully quantify
and predict the visibility impairment caused by aerosol particles in the urban environment. As we have seen from Fig.6 and Fig. 8(b), the hygroscopicity variation is very large in this size range. From our calculation (pie chart in Fig. 8(d)), it would bring about 40% of the total difference between the clean and pollution period, which is similar to the contribution of added aerosol loading.  It's to be noted that all these three factors are coupled together to influence the haze formation, and the hygroscopicity variation would amplify the effect of added aerosol loading and increased ambient RH.

**5. Conclusions**

Submicron particles larger than 300 nm dominate the aerosol light extinction and mass concentration, and constitute a large part of the aerosol surface concentration. Under the exposure to high RH, all these parameters will be enhanced with the addition of water, and then greatly influence the particle mass, visibility degradation, particle chemistry and haze formation.



However, field measurements on aerosol hygroscopicity in previous studies are focused below the size of 350 nm. For larger
particles, detailed description of hygroscopicity characteristics and variations are scarce. Some studies tried to derive the
hygroscopicity of a larger size range from other indirect methods like chemical composition or based on assumptions that
accumulation mode particles share the same hygroscopicity. Till now, no direct evidence has been provided to support these
assumptions or parameterization schemes.

In this study, a comprehensive aerosol field measurement focusing on hygroscopicity properties and variations over extended
size range (50-600 nm) was conducted at a Beijing urban site in winter 2019. During the measurement, an HTDMA instrument
was employed to measure hygroscopic growth factors of particles with dry diameter of 50, 100, 200, 300, 400, 500, and 600
nm at 85% RH from 27 November 2019 to 14 January 2020. This in-situ field measurement of atmospheric aerosols is
representative of urban environment and of great importance to better understand the frequent low visibility events over North
China Plain.

The measurement results show that the mean growth factor (GF) values for the sizes above were 1.13±0.07, 1.25±0.06,
1.31±0.07, 1.30±0.09, 1.24±0.13, 1.16±0.15, 1.12±0.15, respectively. The average GF-PDF present a bi-modal structure
including a less hygroscopic group and a more hygroscopic group. For the particles larger than 300 nm, there is a large fraction
of nearly-hydrophobic or less hygroscopic particles, which decrease the bulk hygroscopicity in this size range. However,
during the polluted episode when the aerosol particles are fully aged, the ensemble mean hygroscopicity of larger particles will
be enhanced significantly because of the growth and expansion of MH group to this size range. Our measurements give a direct
evidence supporting that the variation of MH group brings the biggest change and uncertainty to the bulk hygroscopicity of
particles larger than 300 nm. Our results are consistent with previous HTDMA studies in overlap region (P. F. Liu, Zhao,
Gobel, et al., 2011; Massling et al., 2009; Wang et al., 2018) and the chemical composition studies during the pollution
development (Guo et al., 2014; Wu et al., 2018; Zheng et al., 2016)

The significant variation of larger particle hygroscopicity will contribute much to the variation of particle optical properties
like light extinction coefficient. Size-resolved light extinction coefficients under different pollution conditions are calculated
using Mie scattering theory. From our results, when coupled with increased RH, the influence of κ variation on light
degradation could compete with the added aerosol loading in the haze development. Based on these results, we strongly
recommend simultaneous measurement of hygroscopic properties over size range larger than 300 nm in the future urban
aerosol field studies.

**Competing interests.** The authors declare that they have no conflict of interest.
**Data availability.** The data used in this study can be obtained from this link:
https://pan.baidu.com/s/1Sv9Zi3SJjBf0vhMRH2NNRA.  It is also available when requesting the authors.



**Author contributions.** Chuanyang Shen, Gang Zhao, Weilun, Zhao and Chunsheng Zhao discussed the results; Ping Tian
offered his help in the experiment; Chuanyang Shen wrote the manuscript.

**Acknowledgements.** This work is supported by the National Key Research and Development Program of China
(2016YFC020000: Task 5) and the National Natural Science Foundation of China (41590872).

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





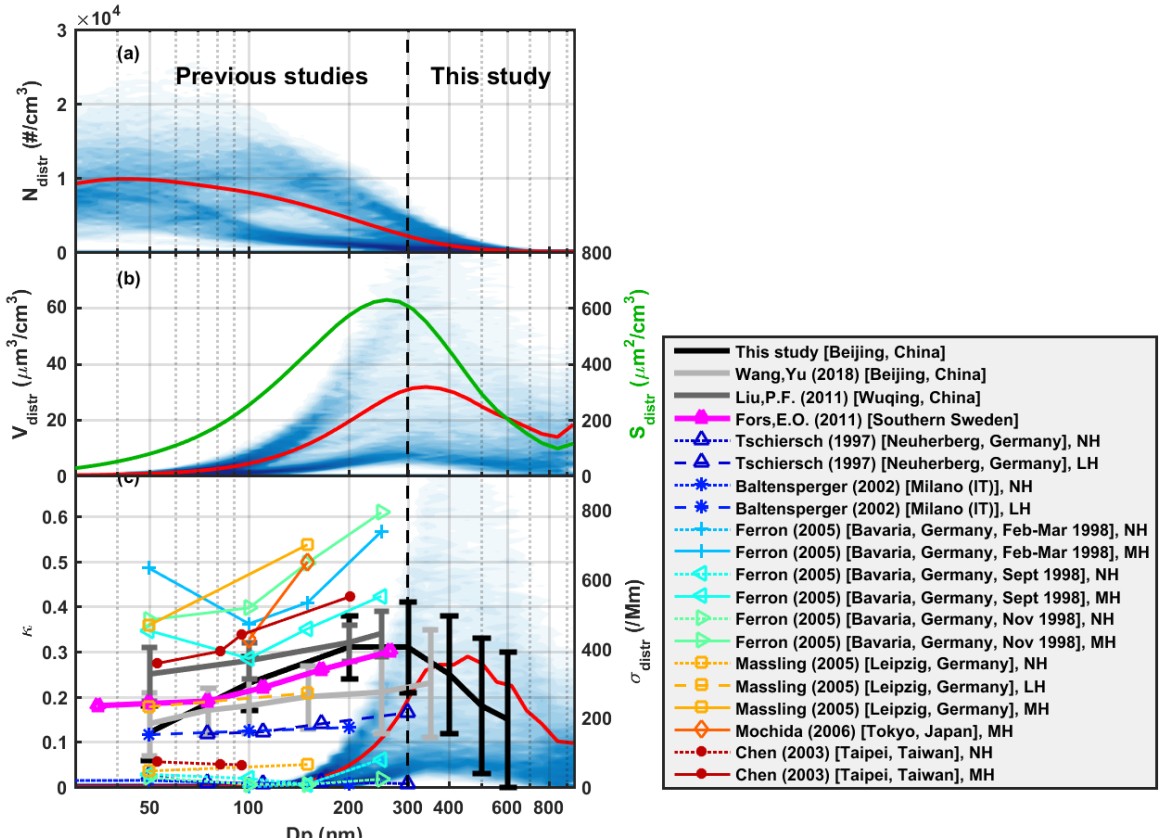

**Figure 1:** Frequency distribution of (a) aerosol number size distribution during the whole measurement period, (b) corresponding volume size distribution and (c) size-resolved light extinction coefficient contribution calculated from Mie theory. The red lines in (a-c) represent the mean values. The lines with marks in Fig. 1(c) denote the hygroscopicity measurement by HTDMA in previous and our studies.





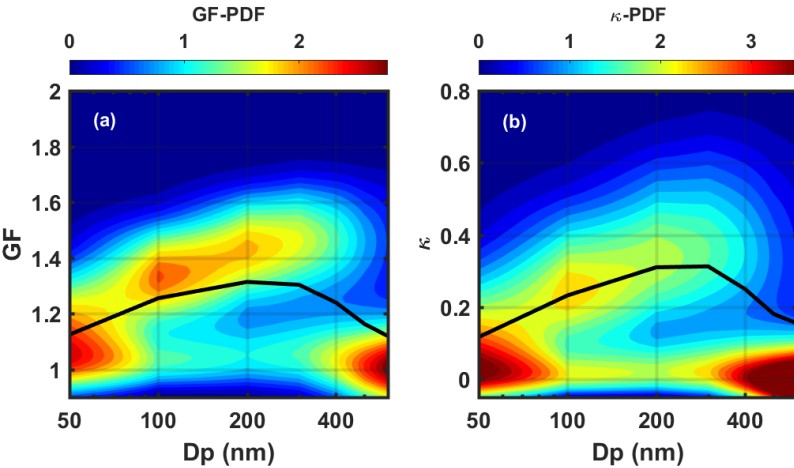

**Figure 2: Averaged probability distribution functions of (a) growth factor and (b) hygroscopicity parameter κ. The black lines in these two figures represent the ensemble mean values.**

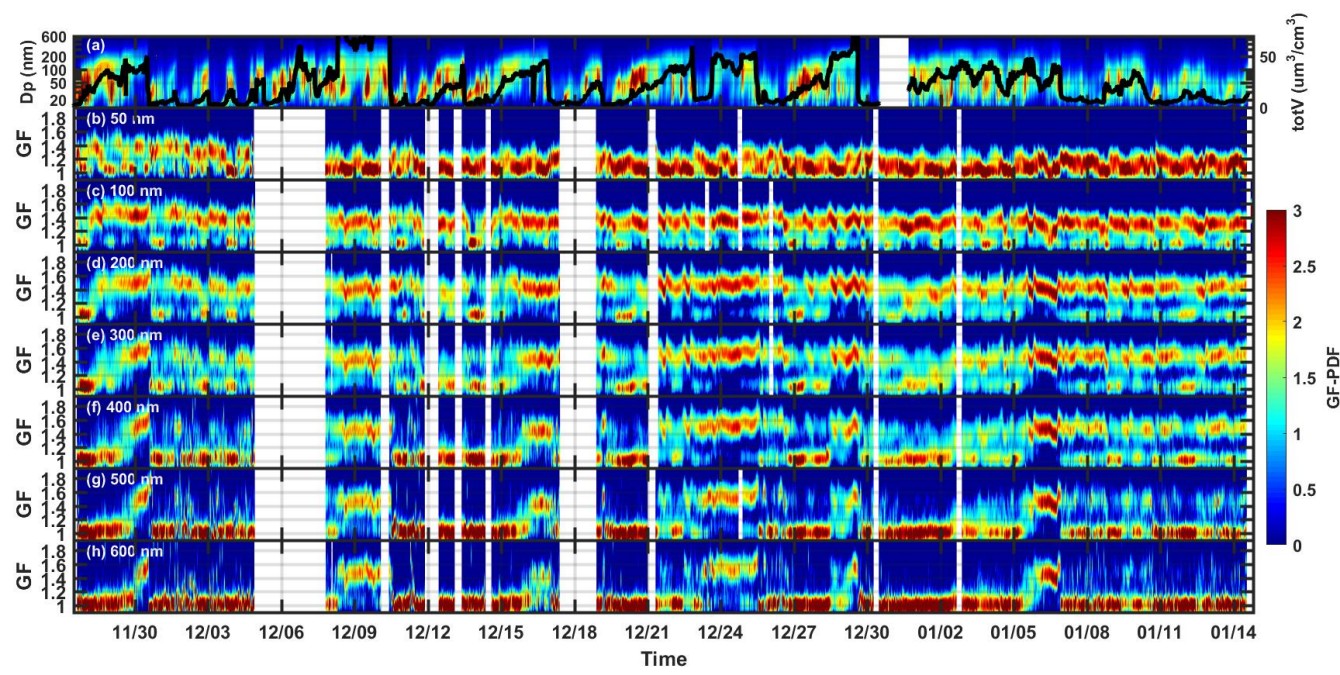

**Figure 3: Temporal evolutions of (a) particle number size distribution and integrated volume concentration (black line), (b-h) growth factor probability distributions over different sizes from 50 nm to 600 nm. The colour contour in (a) denotes the particle number**





concentration (dn/dlogDp ($cm^{-3}$)), in (b-h) denotes the growth factor probability distribution function. Y-axis of (a) is particle diameter in log-scale.


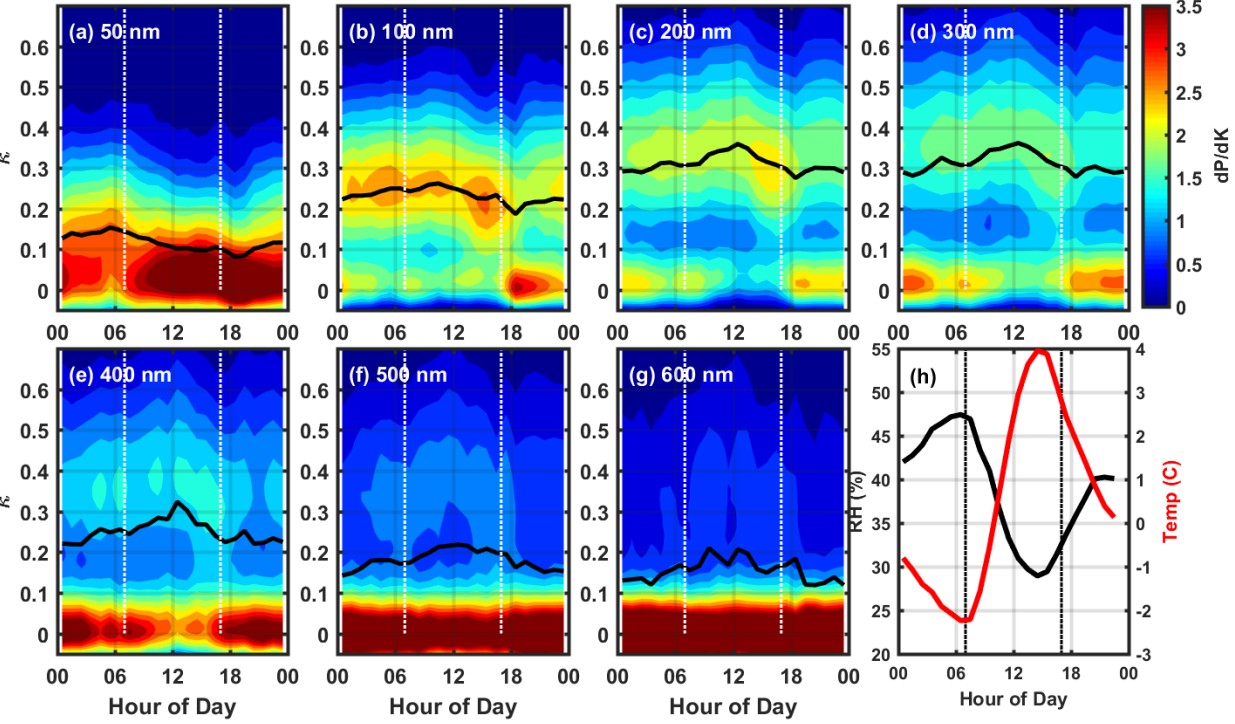

**Figure 4: Diurnal variations of aerosol hygroscopicity distribution function over different sizes from 50 nm to 600 nm. Panel (h) shows the mean diurnal temperature and relative humidity. The black lines in (a-g) are the mean hygroscopicity calculated from the average κ-PDF. The vertical dotted lines in all the figures denote the time of sunrise (~7.00 a.m.) and sunset (~5.00 p.m.).**






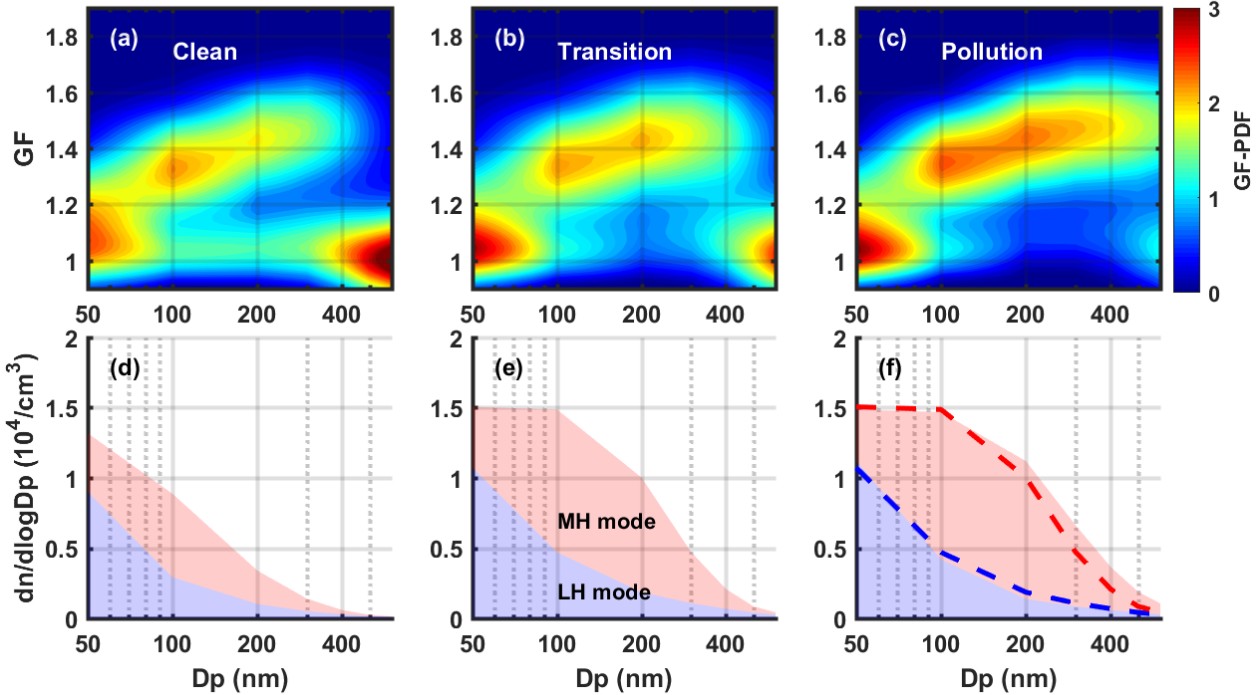

**Figure 5:** (a-c) are the growth factor probability distribution under different pollution conditions. (d-f) are the corresponding mean number size distribution for LH and MH mode. For comparison, we also mark the mean number size distribution under transition stage in dashed lines in panel (f).





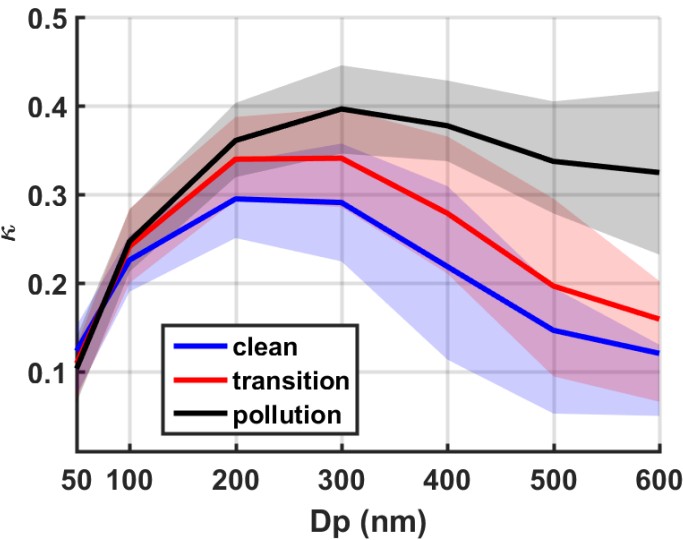

**Figure 6: Mean size-resolved κ under different pollution conditions. The upper and lower limit of semi-transparent patch is the 25% and 75% percentile.**

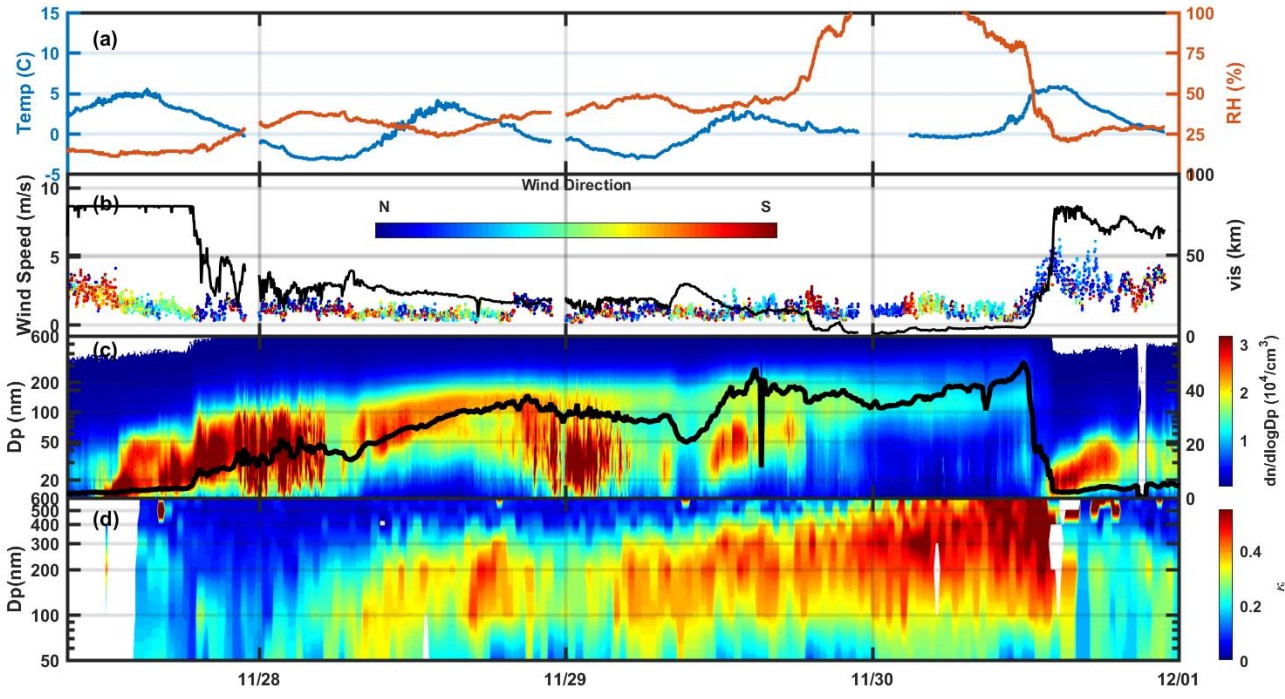

**Figure 7: A pollution cycle from 11/27/2019 to 12/01/2019. (a) is the temporal evolution of temperature (blue line) and RH (orange**
**line); (b) is the time series of visibility (black line) and wind speed. The colour of the points denotes the direction of wind. (c) stands**


for the time series of particle number size distribution and the black line is the integrated volume concentration. Left axis is the log scale of particle diameters and right axis is for the volume concentration. (d) is the size-resolved hygroscopicity parameter κ.

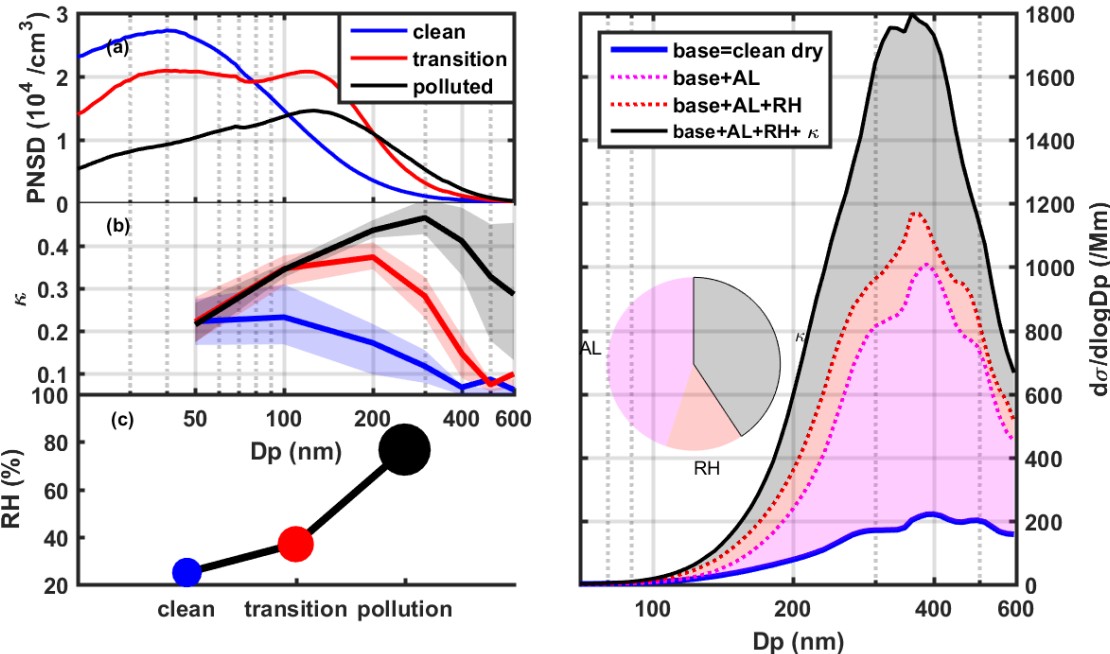

Figure 8: (a) is the mean PNSD under different pollution conditions. (b) is the mean size-resolved κ under different pollution conditions. (c) is the mean RH under different conditions. (d) shows the effects of different factors (aerosol loading, RH and κ variation) to the size-resolved light extinction. The areas of different colors are shown in the pie chart on the left.


**Table 1:** Summary of previous HTDMA measurement of urban aerosol particles.

| Reference | Site | time | Size range (nm) |
|---|---|---|---|
| *Zhang et al. (1993)* | Claremont, California, USA | Summer  1987 | 50-500 |
| *Tschiersch et al. (1997)* | Neuherberg, Germany | Jan-Feb  1997 | 50-300 |
| *Baltensperger et al. (2002)* | Milano (IT) | June-July  1998 | 20-200 |
| *Ferron et al.(2005)* | Bavaria, Germany | Feb-Mar 1998 | 50-250 |
| *Cocker et al. (2001)* | Pasadena, California (Husar et al.) | Aug–Sept 1999 | 50-150 |



| | | | |
|---|---|---|---|
| *Massling et al. (2005)* | Leipzig, Germany | May–Aug 2000 | 50-150 |
| *Mochida et al. (2006)* | Tokyo, Japan | Nov 2000 | 80-150 |
| *Chen et al. (2003)* | Taipei, Taiwan | Oct–Dec 2001 | 53-202 |
| *Zieger et al. (2014)* | Central Europe | Feb-Mar 2009 | 50-265 |
| *Liu et al, (2011)* | Wuqing, China | 7.17-8.12 2009 | 50-250 |
| *Fors et al. (2011)* | South Sweden | May 2008-July 2010 | 35-265 |
| *Holmgren et al. (2014)* | Central France | Sep 2008-Dec 2012 | 25-165 |
| *Y Wang et al. (2018)* | Beijing,China | May 2014-Jan 2015 | 50-350 |


**Table 2:** Summary of the HTDMA measurements of hygroscopic growth at 85% RH.

| *Size (nm)* | *50* | *100* | *200* | *300* | *400* | *500* | *600* |
|---|---|---|---|---|---|---|---|
| *Number of scans* | 1654 | 1680 | 1673 | 1699 | 1710 | 1738 | 1732 |
| *Less-Hygroscopic(GF<1.2)* | | | | | | | |
| *Number fraction* | 0.70±0.19 | 0.35±0.14 | 0.29±0.16 | 0.36±0.20 | 0.50±0.27 | 0.66±0.30 | 0.78±0.29 |
| *Mean GF* | 1.07±0.03 | 1.08±0.03 | 1.09±0.04 | 1.09±0.05 | 1.07±0.05 | 1.05±0.05 | 1.05±0.05 |
| *Spread of GF* | 0.08±0.01 | 0.07±0.01 | 0.07±0.02 | 0.08±0.02 | 0.08±0.02 | 0.08±0.02 | 0.09±0.02 |
| *Mean κ* | 0.05±0.02 | 0.06±0.02 | 0.06±0.03 | 0.06±0.03 | 0.05±0.03 | 0.04±0.03 | 0.03±0.03 |
| *Spread of κ* | 0.059±0.007 | 0.050±0.008 | 0.046±0.012 | 0.049±0.012 | 0.051±0.012 | 0.051±0.012 | 0.051±0.012 |
| *More-Hygroscopic(GF>1.2)* | | | | | | | |
| *Number fraction* | 0.30±0.19 | 0.65±0.14 | 0.71±0.16 | 0.64±0.20 | 0.50±0.27 | 0.34±0.30 | 0.22±0.29 |
| *Mean GF* | 1.29±0.04 | 1.37±0.04 | 1.44±0.05 | 1.46±0.05 | 1.45±0.07 | 1.43±0.10 | 1.43±0.12 |
| *Spread of GF* | 0.05±0.02 | 0.09±0.02 | 0.10±0.01 | 0.11±0.02 | 0.10±0.03 | 0.08±0.04 | 0.07±0.04 |
| *Mean κ* | 0.25±0.05 | 0.31±0.05 | 0.38±0.05 | 0.40±0.06 | 0.38±0.09 | 0.35±0.12 | 0.36±0.15 |
| *Spread of κ* | 0.06±0.03 | 0.10±0.02 | 0.12±0.02 | 0.13±0.03 | 0.11±0.04 | 0.10±0.05 | 0.09±0.06 |
| *Ensemble of all groups* | | | | | | | |
| *Mean GF* | 1.13±0.07 | 1.25±0.06 | 1.31±0.07 | 1.30±0.09 | 1.24±0.13 | 1.16±0.15 | 1.12±0.15 |
| *Mean κ* | 0.12±0.06 | 0.23±0.06 | 0.31±0.07 | 0.31±0.10 | 0.25±0.13 | 0.18±0.15 | 0.15±0.15 |


**Table 3.** Diurnal variations of aerosol hygroscopic properties for different sizes.

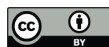


| Size (nm) | 50 | 100 | 200 | 300 | 400 | 500 | 600 |
|---|---|---|---|---|---|---|---|
| Peak (h) | 6 | 10 | 13 | 12 | 13 | 13 | 13 |
| Bottom (h) | 19 | 18 | 19 | 18 | 2 | 0 | 20 |
| **Winter Daytime** | | | | | | | |
| $\kappa_{mean}$ | 0.11±0.07 | 0.24±0.06 | 0.33±0.06 | 0.34±0.09 | 0.27±0.14 | 0.20±0.15 | 0.18±0.17 |
| $nf_{LH-group}$ | 0.72±0.20 | 0.31±0.13 | 0.24±0.13 | 0.31±0.18 | 0.45±0.27 | 0.62±0.31 | 0.73±0.32 |
| $\kappa_{LH-group}$ | 0.05±0.02 | 0.06±0.02 | 0.07±0.03 | 0.06±0.03 | 0.05±0.03 | 0.04±0.04 | 0.04±0.04 |
| $\kappa_{MH-group}$ | 0.24±0.05 | 0.31±0.05 | 0.38±0.05 | 0.41±0.06 | 0.39±0.10 | 0.37±0.14 | 0.37±0.15 |
| **Winter Nighttime** | | | | | | | |
| $\kappa_{mean}$ | 0.12±0.06 | 0.22±0.06 | 0.30±0.08 | 0.30±0.10 | 0.23±0.13 | 0.17±0.14 | 0.14±0.14 |
| $nf_{LH-group}$ | 0.68±0.18 | 0.37±0.15 | 0.32±0.17 | 0.39±0.21 | 0.53±0.28 | 0.69±0.29 | 0.80±0.27 |
| $\kappa_{LH-group}$ | 0.05±0.02 | 0.05±0.02 | 0.06±0.03 | 0.06±0.03 | 0.04±0.03 | 0.04±0.03 | 0.03±0.03 |
| $\kappa_{MH-group}$ | 0.26±0.05 | 0.31±0.05 | 0.37±0.05 | 0.39±0.07 | 0.37±0.08 | 0.35±0.11 | 0.35±0.15 |

**Table 4.** Statistical results of aerosol hygroscopic properties under different pollution conditions.

| Size (nm) | 50 | 100 | 200 | 300 | 400 | 500 | 600 |
|---|---|---|---|---|---|---|---|
| **Clean** | | | | | | | |
| $nf_{LH}$ | 0.68±0.20 | 0.36±0.14 | 0.33±0.16 | 0.41±0.20 | 0.58±0.26 | 0.75±0.26 | 0.86±0.22 |
| Mean GF | 1.13±0.07 | 1.25±0.06 | 1.30±0.07 | 1.28±0.09 | 1.21±0.13 | 1.23±0.13 | 1.08±0.12 |
| Mean $\kappa$ | 0.12±0.07 | 0.23±0.06 | 0.29±0.07 | 0.29±0.10 | 0.22±0.13 | 0.15±0.14 | 0.12±0.14 |
| **Transition** | | | | | | | |
| $nf_{LH}$ | 0.73±0.17 | 0.33±0.15 | 0.21±0.14 | 0.29±0.17 | 0.43±0.25 | 0.61±0.28 | 0.75±0.29 |
| Mean GF | 1.12±0.07 | 1.26±0.06 | 1.34±0.06 | 1.33±0.07 | 1.27±0.11 | 1.19±0.13 | 1.13±0.14 |
| Mean $\kappa$ | 0.11±0.06 | 0.24±0.06 | 0.34±0.07 | 0.34±0.09 | 0.28±0.11 | 0.20±0.13 | 0.16±0.14 |
| **Pollution** | | | | | | | |
| $nf_{LH}$ | 0.74±0.16 | 0.30±0.13 | 0.14±0.11 | 0.16±0.12 | 0.21±0.16 | 0.31±0.22 | 0.39±0.27 |
| Mean GF | 1.11±0.06 | 1.27±0.05 | 1.36±0.05 | 1.38±0.06 | 1.37±0.07 | 1.33±0.10 | 1.30±0.13 |
| Mean $\kappa$ | 0.10±0.05 | 0.25±0.05 | 0.36±0.06 | 0.40±0.07 | 0.38±0.08 | 0.34±0.10 | 0.32±0.14 |