# Peer review of "Measurement report: Aerosol hygroscopic properties extended to 600 nm in the urban environment"

_Atmospheric Chemistry and Physics, 2020_

## Referee Comment (RC1) · Anonymous Referee #1 · 14 Oct 2020

This manuscript investigated the aerosol hygroscopic properties for relatively large particles with a diameter larger than 300nm. As previous studies focus on fine particles, this work presented the significant hygroscopic variation of large particles. This unique measurement can enhance our understanding of aerosol hygroscopic properties. Therefore, I recommend the manuscript for publication in *Atmos. Chem. Phys.,* after properly addressing following technical but important problems.

**Specific Comments:**

1. Line 14: Please define the GF.

2. Line 28-29: Please check the citation format again.

3. Line 48: The citation is wrong, should be (Swietlicki et al. 2008).

4. Section 2.1: I did not understand why the authors introduced the AERONET station here, since it has not been used in this study.

5. Line 131: The GF should be defined before.

6. Line 142: Why the third-moment mean values? Please specify.

7. Line 143: "*... can be calculated from $GF_{mean}$ using the equation above.*" Please specify which equation. The author should number the formula.

8. Line 152: It seems that the BC information is not used in this study. Is there a need to mention this?

9. Section 4.2: The Mie theory was used to describe the aerosol effects on visibility degradation. Please describe to what extent this theory can be used in the estimation.

10. In Fig.7(b): Please accurately replot this panel with the wind direction. I do not understand the color bar, which only shows south wind and north wind.

11. Line 290: It's better to specify the Koschmieder relation here.

12. Line 293: Please at least give some descriptions about the calculation method instead of just putting a citation here. Some necessary but not key information can also be offered in the supplement.

---

## Referee Comment (RC2) · Anonymous Referee #2 · 31 Oct 2020

Review for "Measurement report: Aerosol hygroscopic properties extended to 600 nm in the urban environment" by Shen et al.

The manuscript by Shen et al. report size-resolved aerosol hygroscopicity measurements over an extended size range of 50-600 nm. This size range covers the mode diameter of ambient aerosol particles, and can thus provide more useful information about the optical properties and the climate impact of aerosol hygroscopic growth. They show that on average the number fraction of more hygroscopic mode particles decreases with increasing particle size for 400 nm or larger. However, the more hygroscopic mode in the larger size range is dominant during the polluted events, consistent

with the general consensus that aqueous production of secondary species plays an important role in the formation of winter haze in Beijing. I think the measurement data and analysis are solid. The paper is overall clearly written but could be further improved in English. I, therefore, recommend publication in ACP once the authors address a few issues:

1. In Line 16, the authors mention "unexpected low hygroscopicity"; however, in line 18 they say "this result is supported by previous chemical composition analysis". These two statements seem to be contradictory. Also, the authors should briefly discuss why the hygroscopicity decreases with particle size for large particles, based on previous chemical analysis (inorganic/organic fraction, dust, etc).

2. It might be helpful to explain why the TDMA used in this study can extend the measurements to larger particle sizes. Is this because of a lower sheath flow rate or different geometry of the DMA?

3. What does the red line in Fig. 1c mean?

4. The citation and bibliography styles should be consistent with the ACP format.

---

## Author Comment (AC1) · 14 Dec 2020

Responses to Anonymous Referee #1

General comments:

This manuscript investigated the aerosol hygroscopic properties for relatively large particles with a diameter larger than 300nm. As previous studies focus on fine particles, this work presented the significant hygroscopic variation of large particles. This unique measurement can enhance our understanding of aerosol hygroscopic properties. Therefore, I recommend the manuscript for publication in Atmos. Chem. Phys., after properly addressing following technical but important problems.

General response: Thank you for your thorough and detail review of our manuscript. Your comments are very helpful in improving this work. Here we respond to your comments one by one. Your comments are in italics, and my responses are in plain text.

Specific Comments:

1. *Line 14: Please define the GF.*
   Response: The definition of GF was added in the text.

2. *Line 28-29: Please check the citation format again.*
   Response: The citation and reference was updated to be consistent with ACP format.

3. *Line 48: The citation is wrong, should be (Swietlicki et al. 2008).*
   Response: Thank you for your comment. The correction was made in the new manuscript.

4. *Section 2.1: I did not understand why the authors introduced the AERONET station here, since it has not been used in this study.*
   Response: Thank you for your suggestion. The description concerning the AERONET station was deleted in the new manuscript.

5. *Line 131: The GF should be defined before.*
   Response: It was defined in the abstract.

6. *Line 142: Why the third-moment mean values? Please specify.*
   Response: The third-moment mean value is the volume-weighted mean value of growth factor. It can be directly used to simplify the calculation of total liquid water content for the whole particle population.

7. *Line 143: "... can be calculated from $GF_{mean}$ using the equation above." Please specify which equation. The author should number the formula.*
   Response: Thank you for your comment. The formula was numbered and the equation was also updated in the new manuscript.

8. *Line 152: It seems that the BC information is not used in this study. Is there a need to mention this?*

Response: The BC information was used in the Mie model to calculate the aerosol extinction.

9.  *Section 4.2: The Mie theory was used to describe the aerosol effects on visibility degradation. Please describe to what extent this theory can be used in the estimation.*
    Response: The Mie model is widely employed to estimate the aerosol optical properties given specific physical and chemical characteristics. To evaluate its applicability in aerosol light scattering calculation, an optical closure study was done.
    In the closure study, we have measured aerosol light scattering coefficiencies at (635, 525,450) nm from Aurora 3000 nephelometer. We can also calculate the aerosol light scattering at 525 nm from particle number size distribution (PNSD) using Mie scattering model. The comparison between the measured and calculated aerosol light scattering is shown in the figure below. It can be seen that good consistency is achieved between measured and calculated values, which shows both the data reliability and model applicability.

[Figure]

10. *In Fig.7(b): Please accurately replot this panel with the wind direction. I do not understand the color bar, which only shows south wind and north wind.*
    Response: Thank you for your suggestion. This figure was updated with both wind speeds and directions in the new manuscript.

11. *Line 290: It's better to specify the Koschmieder relation here.*
    Response: The Koschmieder relation was added in the text.

12. *Line 293: Please at least give some descriptions about the calculation method instead of just putting a citation here. Some necessary but not key information can also be offered in the supplement.*
    Response: Thank you for your comment. We added some brief description of the calculation process in the new manuscript.

---

## Author Comment (AC2) · 14 Dec 2020

Responses to Anonymous Referee #2

General Response: Thank you for your positive review of our manuscript. We sincerely appreciate the efforts you have put in the review process, and we improve this work based on your comments and suggestions. Below we will respond to your comments one by one. Your comments are in italics, and my responses are in plain text. All the changes have been included in the newest version of our manuscript.

Specific comments:

*The manuscript by Shen et al. report size-resolved aerosol hygroscopicity measurements over an extended size range of 50-600 nm. This size range covers the mode diameter of ambient aerosol particles, and can thus provide more useful information about the optical properties and the climate impact of aerosol hygroscopic growth. They show that on average the number fraction of more hygroscopic mode particles decreases with increasing particle size for 400 nm or larger. However, the more hygroscopic mode in the larger size range is dominant during the polluted events, consistent with the general consensus that aqueous production of secondary species plays an important role in the formation of winter haze in Beijing. I think the measurement data and analysis are solid. The paper is overall clearly written but could be further improved in English. I, therefore, recommend publication in ACP once the authors address a few issues:*

1. *In Line 16, the authors mention "unexpected low hygroscopicity"; however, in line 18 they say "this result is supported by previous chemical composition analysis". These two statements seem to be contradictory. Also, the authors should briefly discuss why the hygroscopicity decreases with particle size for large particles, based on previous chemical analysis (inorganic/organic fraction, dust, etc).*
   response: Thank you for your comment. These two statements are not contradictory. The "unexpected low hygroscopicity" means that the relatively low average hygroscopicity, especially during clean conditions, is unexpected because the accumulation mode particles used to be thought as more hygroscopic. However, when the pollution evolves, the growth and expansion of MH (more hygroscopic) group particles is consistent with general consensus that secondary species play a dominant role in the pollution evolution. For the second comment, we have some discussions in the section 3.1 (line 175-183). In the discussion, we compare our results with previous chemical analysis and find that the decreased hygroscopicity may be related to the increased mass ratio of EC, dust, or other undefined mass in larger sizes. Since we didn't have simultaneous chemical constituent measurements, we cannot give direct chemical evidence.

2. *It might be helpful to explain why the TDMA used in this study can extend the measurements to larger particle sizes. Is this because of a lower sheath flow rate or different geometry of the DMA?*
   Response: The TDMA can extend the upper size limit because of its unique geometry design. For the DMA, the relationship between electrical mobility size and voltage follows the equation:

$$V = \frac{Q_{sheath} \cdot \ln(\frac{R_{outer}}{R_{inner}})}{2\pi L_{DMA} Z_p}$$

For the same electrical mobility diameter, a lower voltage is needed if the $\frac{R_{outer}}{R_{inner}}$ is smaller. That's to say, a lower ratio of $R_{outer}$ to $R_{inner}$ will extend the upper size limit within the same voltage limit.

We compare the commonly used TSI 3080 DMA with the BMI 2100 DMA in the table below. It can be seen that the special geometry of the BMI 2100 DMA ensures that it can have a larger size limit.

| Model | V | L$_{DMA}$ (cm) | $\frac{R_{outer}}{R_{inner}}$ |
|---|---|---|---|
| TSI 3080 | 10000 | 44.37 | $\frac{1.961}{0.937} = 2.09$ |
| BMI 2100 | 5600 | 34.02 | $\frac{3.613\ cm}{3.120\ cm} = 1.16$ |

3. *What does the red line in Fig. 1c mean?*
   Response: the red line in Fig.1c represents the mean values of calculated size-resolved light extinction coefficients during the whole measurement period. Please check the text in red in the following legend:
   *Figure 1: Frequency distribution of (a) aerosol number size distribution during the whole measurement period, (b) corresponding volume size distribution and (c) size-resolved light extinction coefficient contribution calculated from Mie theory. The red lines in (a-c) represent the mean values. The lines with marks in Fig. 1(c) denote the hygroscopicity measurement by HTDMA in previous and our studies.*

4. *The citation and bibliography styles should be consistent with the ACP format.*
   Response: Thank you for your suggestion. We updated the citation and bibliography format in the new manuscript.